# Changes in the Mechanical Properties of the Horizontal Force-Velocity Profile during a Repeated Sprint Test in Professional Soccer Players

**DOI:** 10.3390/ijerph20010704

**Published:** 2022-12-30

**Authors:** Felipe Hermosilla-Palma, Juan Francisco Loro-Ferrer, Pablo Merino-Muñoz, Nicolás Gómez-Álvarez, Alejandro Bustamante-Garrido, Hugo Cerda-Kohler, Moacyr Portes-Junior, Esteban Aedo-Muñoz

**Affiliations:** 1Escuela de Doctorado, Universidad de Las Palmas de Gran Canaria, 35016 Las Palmas de Gran Canaria, Spain; 2Núcleo de Investigación en Ciencias de la Motricidad Humana, Universidad Adventista de Chile, Camino a Tanilvoro km 12, Chillán 3780000, Chile; 3Pedagogía en Educación Física, Facultad de Educación, Universidad Autónoma de Chile, Talca 3460000, Chile; 4Departamento Ciencias Clínicas, Universidad de Las Palmas de Gran Canaria, 35016 Las Palmas de Gran Canaria, Spain; 5Programa de Posgraduación en Educación Física, Universidad Federal de Rio de Janeiro, Rio de Janeiro 21941-599, Brazil; 6Physical Activity, Health and Education Research Group (AFSYE), Physical Education Pedagogy, Universidad Adventista de Chile, Camino a Tanilvoro km 12, Chillán 3780000, Chile; 7Escuela Nacional de Ciencias del Deporte y la Actividad Física, Facultad de Salud, Universidad Santo Tomás, Santiago 8320000, Chile; 8Laboratory of Psychophysiology and Performance in Sports and Combats, Postgraduate Program in Physical Education, School of Physical Education and Sport, Federal University of Rio de Janeiro, Rio de Janeiro 21941-599, Brazil; 9Unidad de Fisiología del Ejercicio, Centro de Innovación, Clínica MEDS, Santiago 7550557, Chile; 10Escuela de Ciencias de la Actividad Física, El Deporte y la Salud, Facultad de Ciencias Médicas, Universidad de Santiago de Chile, Santiago 9170022, Chile; 11Laboratorio de Biomecánica Deportiva, Unidad de Ciencias Aplicadas al Deporte, Instituto Nacional de Deportes, Santiago 7780421, Chile

**Keywords:** fatigue, acceleration, muscle strength, performance

## Abstract

The objective was to analyze the changes in the horizontal force-velocity profile (HFVP) during the execution of repeated sprinting. Methods: Seventeen first-division Chilean soccer players completed a repeated sprint protocol consisting of eight sprints of 30 m with 25-s pauses between repetitions. The behavior of HFVP variables in each attempt was recorded from video recordings and analysis in the MySprint^®^ application. Results: Differences (*p* < 0.05) were found between sprints in the following: time (T), starting from sprint 5 (F = 35.6; η^2^p = 0.69); theoretical maximum speed (V0), starting from sprint 4 (F = 29.3; η^2^p = 0.51); maximum power (PM), starting from sprint 5 (F = 17; η^2^p = 0.52); rate of decrease in force index produced at each step (DRF), starting from sprint 1 (F = 3.20; η^2^p = 0.17); and RF10, starting from sprint 1 (F = 15.5; η^2^p = 0.49). In comparison, F0 and RFpeak did not present any differences (*p* > 0.05). Conclusion: The HFVP variables more sensitive to the effects of fatigue induced by an RSA protocol are those associated with the production of force at high speeds, being V0, DRF, and Pmax, while those that contribute to the generation of force at the beginning of the sprint, F0 and RFpeak, do not present essential variations.

## 1. Introduction

The ability to perform high-intensity efforts in short periods has been described as one of the main determinants of performance in most sports specialties [1]. In soccer, accelerations, decelerations, changes of direction, and high/very high-intensity runs are most prevalent prior to scoring a goal, becoming determinants of the game’s success [2]. Within these sprints, those races performed at a speed range of 19.1 km/h [3] or over 25.2 km/h [4] comply with the described precept. Given soccer’s characteristics and physical demands, the ability to repeat this type of effort over time is vital. Thus, the ability to repeat sprints (RSA), corresponding to the ability to perform maximal or near-maximal actions of up to 10” in duration, interspersed with short periods of active or passive recovery, corresponds to the classic pattern of efforts in most team sports [5]. The RSA and the ability to cover longer distances by sprinting are linked to the outcome of the match in different leagues [6,7,8], thus becoming one of the fundamental purposes to be developed by coaches and technical teams.

There has been a significant increase in research about the factors/variables affected by fatigue (i.e., repeated-sprint exercise-induced reduction in the maximal power output, force, or speed even though the task can be sustained [5]), during RSA tests, which closely correspond to team-sport activity patterns. For example, Brocherie et al., 2015 [9] showed that a repeated anaerobic sprint test leads to substantial alterations in stride mechanics and leg-spring behavior in professional football players. It is suggested that RSA is also influenced by standard metrics such as step frequency, contact time, and stride time. Besides, whole-body kinematic patterns demonstrate effects on running mechanics that could be determinants for the production/maintenance of repeated sprint performance during an RSA test in team sports athletes [10]. Finally, Van den Tillaar, in 2018 [11], analyzed running kinematics in repeated sprints in training, showing that fatigue induced in repeated 30-m sprints in female soccer players resulted in decreased step frequency and increased contact time. However, the effect of repeated sprints on kinetic variables, such as the horizontal force applied during the sprint, is lacking.

Since maximum speeds are rarely reached in game situations, accelerations assume a preponderant role in the physical performance of athletes [12]. Thus, the analysis of their behavior provides relevant information regarding the components that determine sprinting, for example, the effectiveness of the force applied to the floor and how this allows horizontal displacement, observing that the greater the net force applied horizontally to the floor, the greater the acceleration [13]. Therefore, monitoring and developing variables that determine force production and sprint speed are considered one of the pillars of the training process [14].

An examination of the macroscopic components underlying the force-velocity relationship that explain this performance can be determined using valid and reliable field tests [13,15]. In this sense, the mechanical variables of the horizontal force-velocity profile (HFVP) are as follows: theoretical maximum force (F0), theoretical maximum velocity (V0), maximum power (Pmax), rate of decrease of force produced in each stride (DRF), and maximum value in the force index (RFpeak), which are key to implementing individualized training programs [16], accounting for the mechanical efficiency of the athlete when applying horizontal force, and allow identifying differences depending on the type of sport and the level of sport specialization [17,18]. Likewise, two of its component variables (DRF corresponding to the athlete’s ability to maintain high levels of horizontal force production as the speed of displacement increases, and RFpeak, which represents the percentage of the total force generated that is applied horizontally), are strongly correlated with sprint performance [19].

Furthermore, maintaining performance during repeated sprinting by delaying the associated fatigue is of great interest for performance in soccer [20,21]. However, the study of the horizontal force-velocity profile during tests involving sprint repetition or during acute fatigue in soccer has not been sufficiently explored. In this regard, Nagahara 2016 [14] described how fatigue generated during a soccer match produces impairment in velocity (V0) and in the ability to maintain horizontal force levels during a sprint (DRF). Consistent with this, in the context of rugby sevens, research reports the deleterious effects of fatigue on game-specific movement patterns [22], and the prevalence of high-intensity running [23].

Based on the above, describing the influence of fatigue on the mechanical patterns of sprinting to provide background information to generate specific interventions by the technical bodies becomes necessary. Thus, the purpose of the present study was to analyze the changes in HFVP during repeated sprint execution in male professional soccer players. It is hypothesized that all the variables of the HFVP present a detriment in their expression, with those related to force production at high speeds being the most affected.

## 2. Materials and Methods

### 2.1. Design

A cross-sectional study design was implemented to examine the changes in the horizontal force-velocity profile using time (s), F0 (N/kg), V0 (m/s), peak power (W/kg), DRF (%), RF 10 m (%), and RFpeak (%) during a repeated-sprint ability (RSA) test.

### 2.2. Subjects

Seventeen professional soccer players (age 23.5 ± 5.0 years, body mass 77.1 ± 7.3 kg, height 1.74 ± 0.03 m) belonging to the first division of the Chilean national championship participated. According to reports from the club’s medical team, all players were in the pre-competitive period and free of musculoskeletal pain or injury. They were also informed about the possible risks and benefits of the study, agreeing to participate by signing a voluntary informed consent. The research was approved by the Chilean Adventist University Ethics Committee No. 2022-34. All procedures followed the principles described in the Declaration of Helsinki for human studies [24].

### 2.3. Procedures

The evaluations were carried out during the second week of the pre-competitive period, 4 weeks before the start of the competition, at 9:30 a.m. on the third day of the planned microcycle. All participants performed the tests with soccer shoes on a natural grass field. Participants were instructed to arrive at the performance tests rested, fasting for at least 3 h, and adequately hydrated. Light training sessions were scheduled with the technical team 2 days prior to the evaluation.

### 2.4. Preparation

Prior to the execution of the RSA test, each player completed a 15 min standardized warm-up [25], composed of joint mobility and dynamic stretching, followed by low-intensity aerobic running to conclude with three progressive sprints up to 95% of self-perceived effort, all under the supervision of the team’s physical trainer.

### 2.5. Repeated Sprint Ability (RSA)

After the warm-up, each player performed the RSA test consisting of the execution of eight sprints in a straight line over a distance of 30 m, with a 25 s pause between sprints [26]. According to Girard et al. (2011) [5], the tests that assess this capacity to repeat sprint should be between 10 and 60 s for effort and rest, respectively. The start and finish lines were demarcated on the floor; participants were verbally encouraged to perform each repetition with maximum intensity. Each attempt was recorded by an IPad 8th generation device (Apple, Inc., Cupertino, CA, USA), which was placed perpendicularly 10 m from the surface intended for the race on a tripod at the height of 1.5 m. The analysis of the behavior of the variables of the horizontal velocity force profile were as follows: time (s), F0 (N), V0 (m/s), peak power (W/kg), DRF (%), FV (%), RF 10m (%), and RFpeak (%), which were performed with the MySprint^®^ app, which counts the partial times every 5 m (5 m–10 m–15 m–20 m and 25 m). This app is valid and reliable concerning the reference systems (radar gun and timing photocells) [27].

### 2.6. Statistical Analysis

Descriptive statistics are presented as mean and standard deviation (SD) because the normality assumption was checked through the Shapiro-Wilk test. The inter-sprint comparison was performed with the repeated measures ANOVA test, and the Greenhouse-Geisser sphericity correction was used when the sphericity assumption was not met. Bonferroni post-hoc tests were performed, and effect sizes are presented as partial eta squared (η^2^p). The percentage of change was calculated as a practical fatigue index, where the best value was used as a basal reference and the last attempt, since the first attempt is not always the best value [5], resulting in the following equation: (best value-last value)/best value * ±100 [28]. Finally, the effect size between consecutive sprints was calculated using Cohen’s d average [29], using the following thresholds for qualitative classification: trivial (<0.2), small (0.21–0.6), moderate (0.6 1–1.2), large (1.21–2), very large (2.1–4) [30]. Sample size (n = 17) allows the detection of an effect size of 0.4 in V0 (m/s) in paired data with an alpha error of 0.05 and a power of 80%. All analyses were performed in SPSS^®^ v.28 software with an alpha of 0.05

## 3. Results

The descriptive statistics of the mechanical variables in each sprint of the RSA test are shown in Table 1. Differences (*p* < 0.05) between sprints were found in the following: T (F = 35.6; η^2^p = 0.69; *p* = 0.000), the first sprint presents differences with the remaining seven attempts; V0 (F = 29.3; η^2^p = 0.51; *p* = 0.000), the first sprint presents differences with the third attempt; PM (F = 17; η^2^p = 0. 52; *p* = 0.000), presents differences with sprint 2, 4, 5, 6, 7, and 8; DRF (F = 3.20; η^2^p = 0.17; *p* = 0.047), presents differences with sprint 6, 7, and 8; and RF10 (F = 15.5; η^2^p = 0.49; *p* = 0.000), presents differences with sprint 4, 5, 6, 7, and 8. While for F0 and RFpeak, there were no differences (*p* > 0.05). These differences can be seen in the post-hoc analyses shown in Figure 1.

## 4. Discussion

The objective of this study was to analyze the changes in PFVH during repeated sprint execution in male professional soccer players. Therefore, knowing the impact that RSA has on the different strategies used by athletes to maintain the same HFVP power in different actions becomes preponderant for technical and scientific sections of the sport since it allows for improving the prescription of physical exercise, focused on a better understanding the mechanical properties of the neuromuscular system to optimize performance [31].

Our results show that most of the HFVP variables analyzed decrease their performance in the RSA, except for F0, RF 10 m, and RFpeak. These findings are consistent with what was described by Jiménez-Reyes et al. (2019) [25]. They examined the behavior of the HFVP against the application of repeated sprints in rugby sevens players, demonstrating that the variables associated with force production at high speeds, V0 and DRF, decrease because of fatigue inherent to RSA. However, the ability to produce force at low speeds in F0 and peak RF was not compromised. Furthermore, it is possible to infer that the acute effects of fatigue on the mechanical properties of the neuromuscular system mainly affect those aspects related to the maintenance of high levels of force over time, limiting its expression as movement speed increases.

Our study shows no changes in force expression at low speed, arguing that fatigue does not have a counterproductive effect on performance at the beginning of this type of effort. These findings agree with what was described by Nagahara et al. (2016) [14], who deepened studies on the effect of the efforts made in context on these muscular properties in university soccer players. Therefore, a soccer match could alter this ability to produce force at high speeds (V0), impairing horizontal force production. In the same way, the pattern of the effort made during the game is directly proportional to the loss of this effort, with those athletes who covered the greatest distance presenting the most significant loss associated with V0. Also, our findings show an increase in F0, suggesting a possible immunity of this variable to the effects of fatigue. This last aspect can be partially refuted because some modifications in the technical pattern of athletes have been identified to counteract the counterproductive effects. Wdowski et al. (2020) [32] investigated the effect of a specific resistance stimulus on the kinetics of first support during an acceleration run in professional soccer players. Their findings describe that soccer players modify the movement pattern, decreasing the medial-lateral load and increasing the force in the anteroposterior direction, to counteract the effects generated by fatigue. Our results corroborate this, which shows a decrease in performance in the last 20 m of a 30 m sprint.

From what has been described, it is interpreted that the acute effects of fatigue influence HFVP variables related to the lower zone of the F-V curve, negatively impacting V0 and DRF. As mentioned above, the evidence in this regard is limited. The acute effects of an RSA protocol on sprint kinetic and kinematic variables were recently investigated, finding significant decreases for V0 (ES −1.99, large decrease), Pmax (ES −0.65, large decrease), RFpeak (ES −0.94 large decrease), F0 (ES −0.65 moderate decrease), DRF (ES −0.71 moderate decrease), and concluding that the variable mostly affected is power (Pmax), mainly due to a decrease in velocity (V0) [33].

Besides, the long-term effects of stimuli applied during complete soccer seasons on the variables that make up the HFVP have been described [34]. It has been found that the HFVP components that most decrease their performance at the end of the competition compared to the preseason are those that contribute to the generation of force at low speeds, F0 and RFpeak, while those linked to the maintenance of force at high speeds, V0, DRF, and Pmax, increase. Haugen (2018) [35] analyzed the changes in mechanical variables in the sprint and in others associated with the production of force per unit of time in the vertical jump produced before, during, and at the end of a season of training. Thus, F0, V0, and Pmax increase their expression at the end of the season, with greater changes regarding the previous period and during the season. It is essential to mention that, in both cases, no specific training interventions were carried out aimed at improving the components of the HFVP.

The development of fatigue in repeated sprint efforts appears after the first repetition [36], determined by neural factors and metabolite accumulation [5]. Edouard et al. (2018) [37], analyzed kinetic, kinematic, and electromyography variables before, during, and after a repeated sprint protocol of 12 6-s sprints with a 44-s pause between them. They found a decrease in Vmax and Pmax in the first sprint, as in the present study. In addition, they found decreases in horizontal and vertical force, which could be due to two aspects: (a) the method used in the evaluation (treadmill) [38]; and (b) the number of sprints during the protocol, which could have produced performance fatigability [39], inferring that changes in movement strategies possibly occur, affecting the mechanical variables of sprinting. This situation is corroborated by Romero et al. (2022) [33], who report variations in the knee and hip angulations from the performance of a repeated sprint protocol, presenting different responses between individuals (increases or decreases in joint angle), described as protective of movements injuries.

Also, it has been found that speed decreases proportionally to the number of sprints performed [40]. Similarly, speed losses in the competition are around 7.9% during the last 15 min of a match [41]. Based on this characterization, RSA tests should be proposed to simulate such a scenario to provide a valid parameter for comparison. Although our findings are below these values (percentage change for a time in 30 m = 7.3%), they are not unrelated to those described in the literature, where there is a broad spectrum of reported decreases, ranging from 3.2% to 9.5% [42]. Regarding the sprint durations, our results present a mean of 4.3 ± 0.02 s, adjusting to the described conditions.

Some limitations of our study are that the results presented show the mechanical characteristics of the players during a sprint in preparation for the World Cup. However, these variables can be modified based on the training status, quality of training, responsiveness to training, and nutrition, among others [43]. Another limitation in our study is related to equipment and frequency sampling because Stalker radar has 46.9 Hz. However, the literature shows a frequency to high velocity of up to 60 Hz in motion capture systems [44]. Finally, estimates of variables related to the force-velocity profile have been criticized, especially those related to power output. It is proposed that treatment of scalar quantities (e.g., power) as vectors is not appropriate in biomechanics, and vector quantities as impulses could be taken into account as causative factors in performance [45].

## 5. Conclusions

The HFVP variables most sensitive to the effects of the accumulation of stresses induced by an RSA protocol are those associated with force production at high speeds (i.e., V0, DRF, and Pmax). Conversely, those that contribute to force generation at the beginning of the sprint (F0 and RFpeak) do not present significant variations.

It is recommended to analyze the individual response of the variables obtained by the HFVP in contexts of fatigue, both acutely and chronically. This allows specific interventions depending on the performances that occur at different times. Furthermore, due to the importance of sprinting in the success of decisive actions in soccer, it is necessary to identify the mechanical components that determine its development to manage, reorient, and/or modify the training stimuli accordingly to enhance the individual player’s performance.

## Figures and Tables

**Figure 1 ijerph-20-00704-f001:**
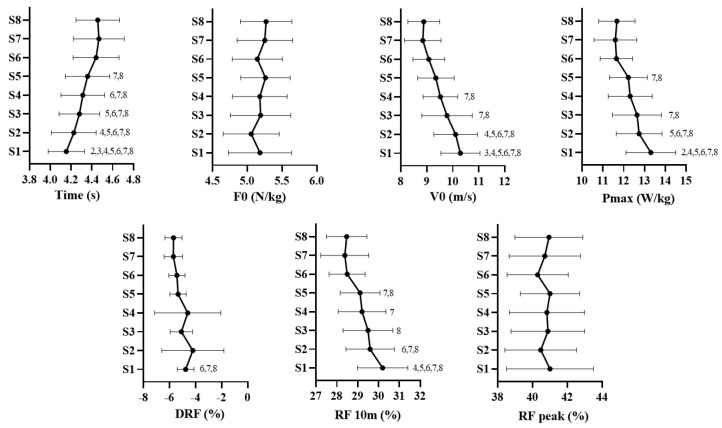
Differences between sprints for the variables of the horizontal force-velocity profile. F0: theoretical maximum force; V0: theoretical maximum speed; Pmax: maximum power; DRF: rate of decrease in force index produced at each step; RF 10 m: force index in the first 10 m; RFpeak: maximum value in force index. 2 = statistically significant difference with sprint number 2; 3 = statistically significant difference with sprint number 3; 4 = statistically significant difference with sprint number 4; 5 = statistically significant difference with sprint number 5; 6 = statistically significant difference with sprint number 6; 7 = statistically significant difference with sprint number 7; 8 = statistically significant difference with sprint number 8.

**Table 1 ijerph-20-00704-t001:** Description and differences for the variables of the horizontal force-velocity profile.

Sprint	T (s)			F0 (N/kg)			V0 (m/s)			PM (w/kg)			DRF (%)			RF 10m (%)			RFpeak (%)		
	M	SD	ES	M	SD	ES	M	SD	ES	M	SD	ES	M	SD	ES	M	SD	ES	M	SD	ES
S1	4.15	0.17	(2) −0.37	5.18	0.46	(2) 0.29	10.29	0.75	(2) −0.23	13.3	1.2	(2) 0.49	−4.77	0.6	(2) −0.37	30.2	1.2	(2) 0.50	41.0	2.5	(2) 0.23
S2	4.23	0.22	(3) −0.26	5.06	0.40	(3) −0.32	10.11	0.84	(3) 0.36	12.7	1.1	(3) 0.09	−4.21	2.4	(3) 0.55	29.6	1.2	(3) 0.09	40.5	2.1	(3) −0.20
S3	4.28	0.20	(4) −0.17	5.19	0.43	(4) 0.03	9.78	0.97	(4) 0.31	12.6	1.2	(4) 0.29	−5.09	0.9	(4) −0.29	29.5	1.2	(4) 0.24	40.9	2.1	(4) 0.03
S4	4.31	0.21	(5) −0.22	5.18	0.39	(5) −0.22	9.52	0.66	(5) 0.25	12.3	1.1	(5) 0.09	−4.61	2.5	(5) 0.47	29.2	1.1	(5) 0.09	40.8	2.2	(5) −0.09
S5	4.36	0.21	(6) −0.38	5.26	0.36	(6) 0.33	9.35	0.70	(6) 0.41	12.2	0.9	(6) 0.68	−5.35	0.6	(6) 0.14	29.1	1.0	(6) 0.69	41.0	1.7	(6) 0.41
S6	4.44	0.22	(7) −0.11	5.14	0.36	(7) −0.28	9.08	0.61	(7) 0.35	11.7	0.8	(7) 0.06	−5.44	0.6	(7) 0.41	28.5	0.9	(7) 0.11	40.3	1.8	(7) −0.22
S7	4.47	0.24	(8) 0.05	5.25	0.40	(8) −0.05	8.85	0.70	(8) −0.05	11.6	1.0	(8) −0.08	−5.71	0.7	(8) −0.02	28.4	1.2	(8) −0.08	40.7	2.1	(8) −0.12
S8	4.46	0.21		5.27	0.37		8.88	0.61		11.7	0.9		−5.69	0.7		28.5	1.0		40.9	2.0	
PC	−7.3			4.23			−13.72			−12.27			−35.2			5.71			0.14		
F	35.62			0.93			29.32			17.02			3.21			15.51			0.45		
*p*	0.000			0.485			0.000			0.000			0.047			0.000			0.868		
η^2^p	0.69			0.06			0.65			0.515			0.17			0.49			0.03		

S: sprint; M: mean; SD: standard deviation; ES: Effect Size; PC: percentage of change; F: F of ANOVA; η2p: partial eta square T: times; F0: theoretical maximum force; V0: theoretical maximum speed; Pmax: maximum power; DRF: rate of decrease of force produced in each stride; RF 10 m: force index in the first 10 m; RFpeak: maximum value in the force index; (2) ES vs. S2; (3) ES vs. S3; (4) ES vs. S4; (5) ES vs. S5; (6) ES vs. S6; (7) ES vs. S7; (8) ES vs. S8.

## Data Availability

Data are available for research purposes upon request to the corresponding author.

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
