# Peer review of "Changes in the Mechanical Properties of the Horizontal Force-Velocity Profile during a Repeated Sprint Test in Professional Soccer Players"

_ijerph, 2022, doi:10.3390/ijerph20010704_

Round 1
Reviewer 1 Report
L29: The objective “was” not “is”
L31: “.” following “m” is redundant
L32-39: Please explain all the abbreviations such as “PFHV, T, V0, PM, DRF, FV, RF10, RFpeak”. After reading the methods and results, I saw the explanations under the table but they must have been provided before that in the abstract as well as in the text where they appear for the first time to make the text more readable.
L87: Again “was” not ”is”
L99: Please put a “.” After the sentence and check the whole text as there are more.
2.5. repeat sprint or repeated sprint?
L106: 9:30 am/pm?
L118: Please use standardized units throughout the text; m or meter
L119: Please check the guidelines for the in-text citations, applicable for the rest of the text.
L119: What capacity? Repeated sprint ability?
L121: “encouraged” seems better than “stimulated”
L122: “recording with” is redundant
L129: Could you provide a reference related to the validity and reliability of the system?
Statistics: Please classify effects size acquired via partial eta square and indicate them in the results section.
Table 1: What is DE? Is it SD? Please correct. Please also indicate effect size (small, medium or large according to partial eta square values)
Discussion: Instead of using “from this” please use conjunctions.
L188-190: The sentence cannot be understood, please revise it.
The discussion is well organized but its language should be checked as well as the rest of the text.
Author Response
Please see the attachment,
Best regards.

Reviewer 2 Report
First of all, I would like to thank the Editor-in-Chief and Associate Editors from International Journal of Environmental Research and Public Health journal for giving me the opportunity to have reviewed the Manuscript ID: ijerph-2115863 titled “Changes in the mechanical properties of the horizontal force-velocity profile during a repeated sprint test in professional soccer players”. The main objective of the present study under review was to analyze the changes in horizontal force-velocity profile (HFVP) during repeated sprint execution in male professional soccer players. There were no working hypothesis presented accompanying the study purpose. A total of 17 professional soccer players participated (Chile first division). The fact that the study was performed with elite athletes should be highlighted. Main outcomes indicated that HFVP variables most sensitive to the effects of fatigue induced by an RSA protocol are those associated with force production at high speeds. Aside from the mentioned strength of the presented work, I have major revisions that need to be addressed by the authors aiming at improve the manuscript as a whole. Please see my specific comments:
P1L33. “Differences (p<0.05) were found between 33 sprints in; T (F=35.6; η²p=0.69), V0 (F=29.3; η²p=0.51), PM (F=17; η²p=0.52), DRF (F=3.20; η²p=0.17), FV (F=8.94; η²p=0.36) and RF10 (F=15.5; η²p=0.49).” - It is important to described in results where the differences appeared (e.g. sprint xx > sprint yy). This could help readers to understand more clearly the outcomes
P2L48. “Under this conceptualization, sprinting, defined as races performed at over 25 km/hr [3]” - Please adjust this sentence in order to avoid state that this value is always used, i.e. there is not an standardisation about sprint thresholds in literature.
P2L49. “…complies with the precept described, representing 3% of the total distance covered in a match [4].” - the same of my previous comment is valid also here. Avoid generalizations as possible. Sprinting distance may be dependent on a range of factors (e.g. competitive level). As such, the referenced 3% value can be not universal. Take care
P2L50 “Given the characteristics and physical demands of soccer, the ability to repeat this type of effort over time is of vital importance. Thus, the ability to repeat sprints (HRS), corresponding to the ability to perform maximum or near-maximum actions of up to 10" in duration, interspersed with short periods of active or passive recovery, corresponds to the classic pattern of efforts in most team sports [5] and becomes one of the purposes to be developed by coaches and technical bodies”- It is interesting to include studies that shows that sprinting/accelerations are determinants to match outcomes in several instances. There is a vast literature dedicated to this topic; please improve the rationale behind the present work regarding this question
P2L63. “Currently, examination of the macroscopic components underlying the force-velocity relationship that explain this performance can be determined by means of valid and reliable field tests [7,9].” Estimates of force and power using similar approach have been criticized. Please see the recent letter by Knudson D. Letter to the editor regarding 'the correlation of force-velocity-power relationship of a whole-body movement with 20 m and 60 m sprint'. Sports Biomech. 2021 Aug 23:1-5. doi: 10.1080/14763141.2021.1968481. Epub ahead of print. It is important to be aware of such recent criticisms and, if pertinent, provide adequate discussion on it.
P2L87. “Thus, the purpose of the present study is to analyze the changes in PFVH during repeated sprint execution in male professional soccer players.” - please insert the hypothesis related to the specific study aim
P2L96. “Seventeen professional soccer players (age 23.5 ± 5.0 years, body mass 77.1 ± 7.3 kg, height 1.74 ± 0.03 m) belonging to the first division of the Chilean national championship”. Add the term “participated” at the end of this sentence. In addition, I urge the authors to insert sample size calculation to ensure that sufficient statistical power was reached in the current number of participants.
P3L129 “This app has been shown to be valid and reliable in relation to the reference systems (radar gun and timing photocells)” - It is mandatory to insert the reference(s) supporting the validity of the tool used in addition to its values
P3L135. If it is difficult to obtain a justifiable number of participants, I suggest to insert post hoc power analysis accompanying p-values. Also inclusion of effect size measures between sprints are advisable.
P3L142. Inclusion of mean and percentage differences together with p-values and effect sizes would increase the practical utility/meaning of the results from the present experiment.
P5 - Figure 1. Please check if “Tiempo (s)” is correct (time?)
Replace “sprint n°” with “sprint number” in the figure legend where necessary
P5L165 “The objective of this study was to analyze the changes in PFVH during repeated sprint execution in male professional soccer players.” - This is possible the main limitation of the current study in terms of originality. At present it is not so evident the advance in relation to existing literature (e.g. Brocherie, F., Millet, G. P., & Girard, O. (2015). Neuro-mechanical and metabolic adjustments to the repeated anaerobic sprint test in professional football players. European Journal of Applied Physiology, 115(5), 891-903. ; de Andrade, V. L., et al. (2021). Critical points of performance in repeated sprint: A kinematic approach. Science & Sports, 36(4), e141-e150 ; Van den Tillaar, R. (2018). Comparison of step-by-step kinematics in repeated 30-m sprints in female soccer players. The Journal of Strength & Conditioning Research, 32(7), 1923-1928.). Please consider the suggested studies (if pertinent) while justifying the relevance of the submitted manuscript.
P7L247 “The HFVP variables most sensitive to the effects of fatigue” - I suggest to have caution to some extent when using the term “fatigue” here as well as across the text ow otherwise provide a clear definition/referenced about this term.
Author Response

(The authors gave the same response as above.)

Round 2
Reviewer 2 Report
First of all, I would like to thank the Editor-in-Chief and Associate Editors from IJERPH for giving me the opportunity to have re-reviewed the Manuscript ID: ijerph-2115863 titled “Changes in the mechanical properties of the horizontal force-velocity profile during a repeated sprint test in professional soccer players”. Some comments of my first review should still be addressed by the authors. If these comments are not properly implemented to the manuscript, it will not be possible to recommend its publication:
P2L75. “Currently, examination of the macroscopic components underlying the force-velocity relationship that explain this performance can be determined by means of valid and reliable field tests [7,9].” Estimates of force and power using similar approach have been criticized. Please see the recent letter by Knudson D. Letter to the editor regarding 'the correlation of force-velocity-power relationship of a whole-body movement with 20 m and 60 m sprint'. Sports Biomech. 2021 Aug 23:1-5. doi: 10.1080/14763141.2021.1968481. Epub ahead of print. It is important to be aware of such recent criticisms and, if pertinent, provide adequate discussion on it. - ALREADY WRITTEN IN THE FIRST REVIEW
P2L110. “Seventeen professional soccer players (age 23.5 ± 5.0 years, body mass 77.1 ± 7.3 kg, height 1.74 ± 0.03 m) belonging to the first division of the Chilean national championship”. I urge the authors to insert sample size calculation to ensure that sufficient statistical power was reached in the current number of participants. - ALREADY WRITTEN IN THE FIRST REVIEW
P9L198. “The objective of this study was to analyze the changes in PFVH during repeated sprint execution in male professional soccer players.” - This is possible the main limitation of the current study in terms of originality. At present it is not so evident the advance in relation to existing literature (e.g. Brocherie, F., Millet, G. P., & Girard, O. (2015). Neuro-mechanical and metabolic adjustments to the repeated anaerobic sprint test in professional football players. European Journal of Applied Physiology, 115(5), 891-903. ; de Andrade, V. L., et al. (2021). Critical points of performance in repeated sprint: A kinematic approach. Science & Sports, 36(4), e141-e150 ; Van den Tillaar, R. (2018). Comparison of step-by-step kinematics in repeated 30-m sprints in female soccer players. The Journal of Strength & Conditioning Research, 32(7), 1923-1928.). Please consider the suggested studies to be included in the portion of literature review of the current manuscript while justifying the relevance of the submitted manuscript. - ALREADY WRITTEN IN THE FIRST REVIEW
P1L41 “Conclusion. the: The HFVP variables more sensitive to the effects of fatigue induced by” - I suggest to have caution to some extent when using the term “fatigue” here as well as across the text ow otherwise provide a clear definition/referenced about this term. - ALREADY WRITTEN IN THE FIRST REVIEW
